# HCV genotype profile in Brazil of mono-infected and HIV co-infected individuals: A survey representative of an entire country

Mariana Fernanda Rodrigues Nutini[1], James Hunter[1], Leila Giron[2], Ana Flavia Nacif Pinto Coelho Pires[3], Igor Massaki Kohiyama[4], Michelle Camargo[1], Maria Cecilia Araripe Sucupira[1], Adele Schwartz Benzaken[5], Paulo Abrão Ferreira[1], Hong-Ha M. Truong[6], Ricardo Sobhie Diaz[1] *

1 Federal University of Sao Paulo, Sao Paulo, Brazil, 2 Wistar Institute–Philadelphia, Pennsylvania, United States of America, 3 Universidade de Brasília, Brasilia, Brazil, 4 Abbott Molecular, São Paulo, Brazil, 5 AIDS Health Care Foundation, Global Program, Sao Paulo, Brazil, 6 Department of Medicine, University of California, San Francisco, California, United States of America

* rsdiaz@catg.com.br

## Abstract

### Introduction

To be eligible for government-provided treatment in Brazil, all HCV-infected individuals are required to be genotyped shortly after diagnosis. We describe the HCV genotype (G) profiles by geographic region, gender, age and HIV co-infection.

### Methods

We assessed 29,071 genotypes collected from HCV-infected individuals from March 2016 to March 2018 (Abbott Real-Time HCV Genotype). We randomly selected 12,336 samples for HIV co-infection testing using an EIA rapid test kit (TR DPP HIV 1/2 Bio-Manguinhos). Descriptive statistical analyses were performed using R.

### Results

Overall, HCV genotype distribution was 40.9% G1A, 30.2% G1B, 23.8% G3, 3.8% G2, 0.7% G4, 0.1% G5 and 0.6% with multiples genotypes. G1A prevalence was 44.4% among males and 35.8% among females. G1B and G2 were more prevalent in older individuals than G1A and G3. G3 was more prevalent in the South region. Of samples tested for HIV co-infection, 15% were HIV+. Median age among HCV/HIV co-infected individuals was 50 years old compared to 57 years old among mono-infected individuals. Distinct HCV genotype prevalence between HCV/HIV co-infected and HCV mono-infected individuals were respectively: G1A 60.6% versus 37.8%, G1B 15.2% versus 32.9%, and G3 18.9% versus 24.7%. G4 was detected among co-infected young men (3.5% versus 0.2% among mono-infected).

**Data Availability Statement:** All relevant data, supporting information files and analytic programs are deposited in a repository entitled HCV

Genotype Profile in Brazil hosted at Open Science Framework (https://osf.io/).

**Funding:** HCV genotype is provided by the Brazilian Ministry of Health. HIV rapid tests were donated to the authors by the Brazilian Ministry of Health. MN was supported by grants from CAPES (Coordenação de Aperfeiçoamento de Pessoal de Nível Superior), Brazil.

**Competing interests:** The authors have declared that no competing interests exist.

## Conclusion

The increasing prevalence of G3, as inferred by the younger ages of the HCV-infected individuals, poses an extra challenge with regards to disease progression. Distinct genotypical profiles between HCV mono-infection and HCV/HIV co-infection warrant future research in order to better understand and help mitigate HCV chains of transmission.

## Introduction

Brazil is a large "continental size" country divided into five macro-regions: North, Northeast, Central, Southeast, and South. Prevalence of HCV infection varies according to different regions among adults ≥18 years old, ranging from 3.2% in the North, to 1.9 in the Central and South, 1.8 in the Southeast, and 1.6 in Northeast. [1] HCV diagnostic and monitoring testing is fully paid for by the Brazilian Public Health system, as is treatment. All citizens 18 years and older are entitled to receive treatment. The Brazilian national protocol for treatment uses the combination sofosbuvir/ledipasvir (patients without previous treatment for G1), sofosbuvir/velpatasvir (patients without previous treatment for non-G1 strains) and glecaprevir/pibrentasvir + sofosbuvir (retreatment). http://conitec.gov.br/images/Relatorios/2018/Relatorio_PCDT_HepatiteC.pdf).

HCV/HIV co-infection is of particular concern because liver fibrosis progression rate in this population is more rapid. [2, 3] HIV viremia is an independent predictor of liver fibrosis progression among these patients. [4]

Contemporary characterization and classification of viral pathogens rely on the genetic profile of these agents according to genome sequencing followed by phylogenetic analysis. Therefore, viral pathogens are usually divided into genotypes or types, with genetic similarities ranging between 66 to 69%, sub-genotypes or subtypes with similarities ranging from 77 to 80%, and the so-called viral quasispecies with similarities ranging from 91 to 99%. [5] The emerging viral pathogen is thought to evolve genetically in the host forming the viruses quasispecies. By founder effect, new diverse viral pathogens may determine the genetic profile of the virus in a given population or geographic region. It is also conceivable that distinct viral types or subtypes, as distinct biologic entities, may cause distinct pathogenicity, transmissibility or treatment response to antivirals. As a classic example, disease progression, response to antiretroviral treatment and transmissibility are distinct between HIV-1 and HIV-2. [6]

It has been demonstrated that HCV is divided into seven genotypes with several sub-genotypes including 1a, 1b, 1c, 2, 3, 4, 5, and 6. [7] Endemic genotypes include genotypes 1 and 4 in Central Africa, 2 in West Africa, 5 and 7 in Central/Southern Africa, and 3 and 6 in the South and East Asia. [8–12] Some subtypes in endemic regions have spread and become established in distinct regions of the globe by founder effect. The epidemic subtypes in the developed world have spread widely after the Second World War as a result of increases in transfusions of blood components, invasive medical and dental procedures, and intravenous drug use. [13–15]

HCV genotype is important since the genetic barrier to resistance to direct antiviral agents (DAAs) is usually lower with genotype 1A compared to 1B. [16] For example, the natural polymorphisms Q80K present in some genotype 1A strains may also relate to decreased response to simeprevir in the presence of cirrhosis. [17, 18] Genotype 3 may harbor natural polymorphisms that may impair response to NS5A antagonists in the presence of cirrhosis. [19] Although pangenotypic DAAs have been developed, such as the combination of sofosbuvir

and daclatasvir, sofosbuvir and velpatasvir, glecaprevir and pibrentasvir, treatment still needs to be adjusted among cirrhotic patients with genotype 3. As well, the duration of treatment using sofosbuvir and daclatasvir or use of ribavirin for sofosbuvir and velpatasvir needs modification. Besides guiding treatment decisions, evaluation of the molecular biology dynamics in a given region can provide clues about the expansion of a given strain, which conceivably may help inform the design of future vaccines.

As of January 2016, all HCV genotype determination for the Brazilian Ministry of Health program is performed at a centralized molecular biology laboratory (Laboratorio Centro de Genomas, São Paulo, Brazil). Individuals with recent HCV diagnoses by serology have an HCV viral load as a confirmatory test and a genotype test, followed by a liver fibrosis evaluation. We present the first reported results of HCV genotype profile among individuals recently diagnosed with HCV in Brazil by geographic region, gender, age and HIV co-infection.

## Methods

Samples from 29,071 HCV recently diagnosed individuals in Brazil collected from March 2016 to March 2018 were analyzed. To assess the prevalence of HCV/HIV co-infection, 12,336 samples were randomly selected for HIV testing. As recommended by the World Health Organization for HIV prevalence surveillance, one EIA test is sufficient if the HIV prevalence in the given population is greater than 10%. [20]

### Ethical statement

This study was approved by the Ethical Committee of the Federal University of São Paulo (#1687/2016). This study was authorized by and performed in partnership with the Sexually Transmitted Infections/AIDS and Viral Hepatitis Department of the Brazilian Ministry of Health. After the initial HCV genotype determination, all data and biological samples from patients were fully anonymized prior to samples being transferred from the Laboratorio Centro de Genomas, São Paulo—Brazil to the Retrovirology Laboratory of the Federal University of Sao Paulo—Brazil for HIV testing and statistical analysis. The Ethical Committee granted this study a waiver of informed consent.

Samples were tested for HCV genotype using the commercial Abbott kit according to the manufacturer's specifications (https://www.abbottmolecular.com/products/realtime-hcv-genotype-II.html). Briefly, the *Abbott Real Time HCV genotype II* is an in vitro RT-PCR assay using polymerase chain reaction for the determination of HCV genotypes in the plasma or serum of individuals with HCV infection with the lower detection limit of 500 I.U. of HCV RNA copies/mL. After an initial PCR amplification using primers designed for HCV conserved region, fluorescent oligonucleotide probes with specific labeling for HCV genetically diverse genomic regions of genotypes 1, 2, 3, 4, 5 and 6, and subtypes 1a and 1b were used for genotype assignment. When more than one distinct fluorescence was detected in a single plasma sample, dedicated software interpreted the result as infections with more than one HCV genotype. During the sample collection period, 3.2% of collected plasma samples did not yield a clear genotype assignment using the *Abbott Real Time HCV genotype II* platform, and these patients were excluded from further analysis.

All samples submitted to HCV subtyping by the Ministry of Health were tested for HIV antibodies using the rapid *HIV test kit TR DPP® HIV1/2, Bio-Manguinhos—Institute of Technology in Immunobiology, Industry*, in accordance with the manufacturer's instructions. The test qualitatively determines the presence of the anti-HIV antibody by immunochromatography, is easy to perform, does not require laboratory infrastructure for its performance and can generate results within 30 minutes. [21]

The data was organized and analyzed using the R Statistical Computing System on a Macintosh (version 3.5.1). [22] Initial data cleaning and graphics used the Tidyverse set of packages for R. [23] Descriptive statistics were performed with the DescTools package and group comparisons conducted by chi-squared analytic algorithms in the base R system and the gmodels package. [24, 25]

### Availability of raw data and files

All relevant data, supporting information files and analytic programs are deposited in a repository entitled HCV Genotype Profile in Brazil hosted at Open Science Framework (https://osf. io/)

## Results

There was a large representation of women in this survey, as 41.3% of the 29,071 individuals genotyped were female. The samples were also reasonably representative of each Brazilian Macro-region. We evaluated 8,012 individuals from the South region (27.6%), 15.767 from the Southeast region (54.2%), 1,128 from Center (3.9%), 2,796 from the Northeast (9.6%) and 1,368 from the North region (4,7%). As seen in Table 1, the current overall prevalence of genotype profile was highest for G1A at 40.9%, followed by G1B at 30.2%, and G3 at 23.8%, with the prevalence of other genotypes at less than 4%.

We explored whether median age correlated with the expansion dynamics of the epidemic to assess the hypothesis that the lower the age, the greater the expansion of a particular genotype or specific status (gender, co-infection, etc.). The overall mean age for men was 54.1 years

**Table 1. Demographic data of HCV mono and HIV co-infected individuals describing HCV genotype according to gender and age.**

| Overall | | | | | | | |
|---|---|---|---|---|---|---|---|
| | Genotypes | | | | | | |
| | G1 A | G1 B | G2 | G3 | G4 | G5 | Multiple |
| (%) | 40,9 | 30,2 | 3,8 | 23,8 | 0,7 | 0,1 | 0,6 |
| median Age (55) | 52 | 60 | 61 | 56 | 45 | 63 | 56 |
| Male | 44,4 | 26,1 | 3,5 | 24,2 | 0,9 | 0,1 | 0,7 |
| median Age Male (54) | 52 | 59 | 60 | 55 | 44 | 71 | 56 |
| Female | 35,8 | 36,0 | 4,1 | 23,1 | 0,3 | 0,2 | 0,6 |
| median Age Female (57) | 53 | 60 | 61 | 57 | 54 | 56 | 54 |
| HCV Mono-infected | | | | | | | |
| (%) | 37,8 | 32,9 | 3,7 | 24,7 | 0,2 | 0,1 | 0,6 |
| median Age (57) | 54 | 61 | 63 | 57 | 56 | 63 | 58 |
| Male | 40,9 | 29,1 | 3,6 | 25,3 | 0,2 | 0,1 | 0,8 |
| median Age Male (56) | 53 | 60 | 62 | 56 | 50 | 71 | 58 |
| Female | 33,5 | 38,0 | 3,9 | 23,9 | 0,2 | 0,2 | 0,5 |
| median Age Female (58) | 55 | 62 | 63 | 58 | 63 | 56 | 63 |
| HIV co-infected | | | | | | | |
| 15,0% prevalence | | | | | | | |
| (%) | 60,6 | 15,2 | 2,2 | 18,9 | 2,5 | 0,0 | 0,6 |
| median Age (50) | 49 | 51 | 54 | 51 | 38 | | 50 |
| Male | 61,7 | 13,7 | 2,3 | 18,1 | 3,5 | 0,0 | 0,7 |
| median Age Male (49) | 49 | 51 | 54 | 50 | 38 | | 52 |
| Female | 57,6 | 19,1 | 1,9 | 20,9 | 0,0 | 0,0 | 0,4 |
| median Age Female (50) | 50 | 52 | 54,5 | 52 | | | 45 |

**Table 2. Difference in mean and median ages by gender according to HCV genotype groupings.**

| Genotype | Males | | Females | |
|---|---|---|---|---|
| | Mean | Median | Mean | Median |
| G1A, G3 & G4 | 52.47 | 53 | 53.97 | 55 |
| G1B, G2, G5 & Multiples Genotypes | 57.91 | 59 | 59.12 | 61 |
| t-test value | -28.284 | | -21.432 | |

old, which was significantly lower than 56.1 years old for women (Welch's Two Sample t-test, t = 13.367, df = 23251, p < $10^{-6}$). The median age for men was 54 and for women was 57. We also grouped the subtypes into those with higher and lower median ages. Among these, the mean age was significantly lower among individuals infected with G1A, G3 and G4 for both genders (p < $10^{-6}$), as shown in Table 2. When considering just G4, the median age for men was 44, which was significantly lower than 54 for women (Welch's Two Sample t-test, t = 2.750, df = 38.5, p = 0.0045).

As shown in Table 3, G1A was more prevalent in the South, Southeast and Central regions, while G1B was more prevalent in the Northern regions (p < $10^{-6}$). Of note, G3 was more

**Table 3. HCV genotype prevalence among the total number of individuals, HCV mono-infected individuals and HCV/HIV co-infected individuals according to gender and Brazilian geographic regions.**

| Overall | | | | | | | |
|---|---|---|---|---|---|---|---|
| | Genotypes | | | | | | |
| | G1 A | G1 B | G2 | G3 | G4 | G5 | Multiple |
| Total | 40.9 | 30.2 | 3.8 | 23.8 | 0.7 | 0.1 | 0.6 |
| (%) males/females | 44.4/35.8 | 26.1/36.0 | 3.5/4.1 | 24.2/23.1 | 0.9/0.3 | 0.1/0.2 | 0.7/0.6 |
| Regions | | | | | | | |
| North | 26.2 | 44.7 | 5.4 | 22.8 | 0.1 | 0.1 | 0.8 |
| Northeast | 31.6 | 42.3 | 2.8 | 22.4 | 0.4 | 0 | 0.5 |
| Central | 50.7 | 25.7 | 2.3 | 19.9 | 0.7 | 0 | 0.7 |
| Southeast | 42.1 | 33.5 | 3 | 19.7 | 1 | 0.1 | 0.5 |
| South | 42.7 | 17.7 | 5.5 | 33 | 0.2 | 0 | 0.9 |
| HCV Mono-infected | 37.8 | 32.9 | 3.72 | 24.7 | 0.19 | 0.09 | 0.6 |
| (%) males/females | 40.9/33.5 | 29.1/38.0 | 3.6/3.9 | 25.3/23.9 | 0.2/0.2 | 0.1/0.2 | 0.8/0.5 |
| Regions | | | | | | | |
| North | 24.6 | 45.6 | 6.3 | 22.8 | 0 | 0.2 | 0.5 |
| Northeast | 30.4 | 43.1 | 2.7 | 23 | 0.2 | 0 | 0.6 |
| Central | 49.1 | 25.4 | 2.2 | 22.6 | 0.4 | 0 | 0.2 |
| Southeast | 38.7 | 36.5 | 3 | 20.7 | 0.3 | 0.2 | 0.7 |
| South | 39.8 | 19.1 | 5.5 | 34.9 | 0 | 0 | 0.8 |
| HIV co-infected | 60.6 | 15.2 | 2.16 | 18.9 | 2.54 | 0 | 0.59 |
| (%) males/females | 61.7/57.6 | 13.7/19.1 | 2.3/1.9 | 18.1/20.9 | 3.5/0 | 0 | 0.7/0.4 |
| Regions | | | | | | | |
| North | 48 | 20 | 0 | 28 | 0 | 0 | 4 |
| Northeast | 46.4 | 34 | 4.1 | 12.4 | 2.1 | 0 | 1 |
| Central | 67.9 | 12.5 | 0 | 17.5 | 1.8 | 0 | 0 |
| Southeast | 61.3 | 17.2 | 1.7 | 15.2 | 4.1 | 0 | 0.5 |
| South | 61.4 | 9 | 2.9 | 25.9 | 0.2 | 0 | 0.6 |

prevalent than G1B in the South region (Chi-squared test, $\chi^2$ = 4059.6, df = 1, p < $10^{-6}$), as this genotype has a similar prevalence as G1 among women in this region (Table 3).

Prevalence of HCV/HIV co-infection was 15.0%. HCV/HIV co-infection prevalence was higher among men compared to women (18.1% versus 10.4%, Chi-squared test, $\chi^2$ = 140.3, df = 1, p < $10^{-6}$), and the mean age of HCV/HIV co-infected individuals was lower compared to HCV mono-infected persons (Welch's Two Sample t-test, t = 28.1, df = 3069, p < $10^{-6}$ (Table 3).

After detection of HIV seropositivity, we cross-checked the Brazilian National Database to confirm whether these patients had been reported to the system. HIV reporting is required by the Brazilian Ministry of Health in order to provide monitoring testing and antiretroviral treatment to the HIV-infected individuals. According to the national data base, 9% of these individuals had not been previously diagnosed with HIV infection.

HCV/HIV co-infection prevalence varied by geographic regions, ranging from 19.1% in the South, to 15.6% in the Southeast, 7.8% in the Northeast, 10.9% in the Central region, and 4.3% in the North (Chi-squared test, $\chi^2$ = 153.6, df = 4, p < $10^{-6}$). As seen in Table 3, HCV/HIV co-infection was more prevalent among individuals with G1A: 60.6% among co-infected and 37.8% among mono-infected (One Sample Proportions test, $\chi^2$ = 1584.4, df = 1, p < $10^{-6}$). It was also lower for individuals with G1B: 15.2% among co-infected compared to 32.9% of mono infected ($\chi^2$ = 1584.4, df = 1, p < $10^{-6}$). G3 was less prevalent among HCV/HIV co-infected individuals (18.9%) than among mono-infected persons (24.7%, $\chi^2$ = 1706.1, df = 1, p < $10^{-6}$) but was still higher than G1B (15.2%) and G2 (2.2%).

The overall median age in this survey was 55 years old and was lower among HCV/HIV co-infected individuals (50 years old) than among HCV mono-infected individuals (57 years old). The comparison of means by a Welch's Two-Sample t-test produced a t-value of 28.147 on 3069 df with a p-value < $10^{-6}$. The median age among men was lower than women in HCV mono-infected individuals, but was similar between genders among HCV/HIV co-infected individuals (Table 2). Also, the median age for individuals with G4 was significantly lower among HCV/HIV co-infected men (38 years old) compared to HCV mono-infected women (50 years old) (Kruskal-Wallis rank sum test, $\chi^2$ = 7.7, df = 1, p = 0.006). There were no HCV/HIV co-infected females with G4 identified in the study.

## Discussion

The prevalence of distinct HCV genotypes in Brazil reveals an interesting dynamic suggesting establishment and expansion of strains considered in the recent past as minority strains in the country. When analyzing the HCV prevalence by region, gender, age and HIV co-infection, one can speculate about trends in the HCV genotype profile, probably as the result of multiple founder effect phenomena.

For example, in contrast to what has been previously described in Brazil [26, 27], G1B was not the dominant genotype detected. Prevalence of G1A was higher overall, with an increasing prevalence of G3 based on the younger ages of individuals with G3 compared to those with G1. Furthermore, the median age among individuals with G1A and G3 was lower than G1B and G2, suggesting that G1A and G3 are expanding in the Brazilian population. Interestingly, there was a higher prevalence of G1A and G3 in the southern regions, such as the Central, Southeast and South regions, compared to a higher prevalence of G1B in the northern regions. Based on these results, we speculate that the HCV epidemic is expanding in the Southern regions of the country, and more attention should be given to the demographic characteristics associated with each genotype profile observed in order to interrupt transmission chains. Of

particular note was the high prevalence of G3 observed among HCV-infected women in the South region of Brazil.

The current data also corroborates a prior study that used a phylogenetic model to infer the timing for the introduction of specific HCV genotypes in Brazil [28]. According to the Time to Most Recent Common Ancestor (TMRCA) as inferred by Bayesian analysis, G1B entered in Brazil around the year 1923 (1844–1967), whereas G3 entered in 1967 (1955–1980), and G1A in 1979 (1967–1987). More interestingly, the inferred Growth rate (r) was 0.26 for G1B, 0.32 for G3 and 0.4 for G1A. These *r* values explain the finding of G1A becoming more prevalent than G1B and suggests that soon, the prevalence of G1B will also be overtaken by G3. It is also conceivable that the once rare HCV G4 is growing extensively among HIV co-infected young men as a result of the founder effect. This result points to the need for a better understanding of the demographic characteristics associated with the mode of HCV transmission, such as among men that have sex with men, in order to interrupt potential chains of transmission.

We also found a high prevalence of HCV/HIV co-infection among recently diagnosed HCV-infected individuals (15%), and HCV/HIV co-infection prevalence higher among men (18.1%) as compared to women (10.4%). The distribution of HIV cases by sex has changed from 1 woman to 28 men in the mid-1980's to 1 woman to 1.5 men in 2017. The increased incidence among women emphasizes that they are becoming more vulnerable to HIV infection in Brazil. [29] Nonetheless, HCV/HIV co-infection is more prevalent among men in this current survey, which concurs with internationally observed trends of high HCV incidence among HIV-infected man who have sex with men. [30, 31] We recognize that due to the nature of the current study, risk factors for HIV acquisition were not collected as part of this survey, thereby precluding more definite conclusions.

We also observed that the median age among HCV/HIV co-infected individuals was lower than among HCV mono-infected individuals, which also suggests a growing epidemic among HIV-infected individuals. Therefore, efforts should target better HCV prevention strategies among the HIV-infected population. In addition, prevalence of HCV/HIV co-infection was higher in the southern regions compared to the North, suggesting a higher vulnerability to HCV infection in the South. Interestingly, Southern regions of Brazil are wealthier and more developed than the North.

The importance of the genetic diversity of a viral pathogen relies on the fact that genetically distinct strains may be considered as distinct biologic entities. For instance, longitudinal studies have demonstrated that G3 may significantly increase the HCV pathogenicity with respect to rates of steatosis, fibrosis, and hepatocellular carcinoma. It has been demonstrated that in comparison to other HCV genotypes, G3 may be associated to higher lipid levels and hepatic steatosis. [32] A higher rate of liver fibrosis over time is also associated to G3, with a 50% increased risk in G3-infected individuals compared to other genotypes. [33] It has also been described that individuals infected with G3 may have a 31% higher risk of developing cirrhosis and 80% higher risk of developing hepatocellular carcinoma in comparison to individuals with G1. [34] In this sense, the expansion of G3 HCV in certain areas of Brazil may constitute a worrisome feature of the HCV epidemic.

Although the treatment response on interferon-based regimens is better among individuals with G3 as compared to other genotypes, treatment with DAAs still poses an additional challenge. [35, 36] In general, G3 is not susceptible to first-generation HCV protease inhibitors such as telaprevir [37] Furthermore, when NS5A inhibitors are used, it has been demonstrated that polymorphisms previously present at the NS5A genomic region would impair the virologic response to treatment among cirrhotic patients. [38]

With regards to genotype 1, it has also been demonstrated that the genetic barrier to resistance to HCV protease inhibitors is lower for G1A compared to G1B, leading to a faster

virologic failure among individuals with G1A. [39] Therefore, an increase in the prevalence of G1A and G3 poses an additional challenge for treatment with some DAAs.

We recognize that the prevalence of HCV/HIV co-infection may have been overestimated if HCV diagnostic testing was requested more frequently by individuals who had been previously diagnosed with HIV. We also recognize that the lack of data on risk factors for HCV and HIV acquisition precludes more definite conclusions and speculations about the HCV genotype profiles described here. Nonetheless, this is the largest report of the distribution of HCV genotype in Brazil and we believe our study sample was representative of the entire country. Our study showed an unprecedent growth of G1A and G3 in certain Brazilian regions and a very high prevalence of HCV/HIV co-infection among individuals recently diagnosed with HCV.

## Acknowledgments

We thanks Fabiana Santos for helping with HIV rapid testing.

## Author Contributions

**Data curation:** Mariana Fernanda Rodrigues Nutini, Leila Giron, Ana Flavia Nacif Pinto Coelho Pires, Igor Massaki Kohiyama, Michelle Camargo, Adele Schwartz Benzaken.

**Formal analysis:** Mariana Fernanda Rodrigues Nutini, James Hunter, Paulo Abrão Ferreira, Hong-Ha M. Truong.

**Funding acquisition:** Ricardo Sobhie Diaz.

**Investigation:** Mariana Fernanda Rodrigues Nutini, Leila Giron, Michelle Camargo, Ricardo Sobhie Diaz.

**Methodology:** James Hunter, Paulo Abrão Ferreira, Ricardo Sobhie Diaz.

**Resources:** Ana Flavia Nacif Pinto Coelho Pires, Adele Schwartz Benzaken.

**Software:** James Hunter.

**Supervision:** Maria Cecilia Araripe Sucupira, Ricardo Sobhie Diaz.

**Visualization:** Ana Flavia Nacif Pinto Coelho Pires, Adele Schwartz Benzaken.

**Writing – original draft:** Ricardo Sobhie Diaz.

**Writing – review & editing:** Hong-Ha M. Truong.

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
