## [Decision Letter · Decision Letter 0]

10 Oct 2019

PONE-D-19-24177

HCV genotype profile in Brazil of mono-infected and HIV co-infected individuals: a survey representative of an entire country

PLOS ONE

Dear Dr. Diaz,

Thank you for submitting your manuscript to PLOS ONE. After careful consideration, we feel that it has merit but does not fully meet PLOS ONE’s publication criteria as it currently stands. Therefore, we invite you to submit a revised version of the manuscript that addresses the points raised during the review process.

We would appreciate receiving your revised manuscript by Nov 24 2019 11:59PM. To enhance the reproducibility of your results, we recommend that if applicable you deposit your laboratory protocols in protocols.io, where a protocol can be assigned its own identifier (DOI) such that it can be cited independently in the future. For instructions see: http://journals.plos.org/plosone/s/submission-guidelines#loc-laboratory-protocols

We look forward to receiving your revised manuscript.

Kind regards,

Jason Blackard, PhD

Academic Editor

PLOS ONE

**Journal Requirements:**

2. In ethics statement in the manuscript and in the online submission form, please provide additional information about the patient records/samples used in your retrospective study. Specifically, please ensure that you have discussed whether all data/samples were fully anonymized before you accessed them and/or whether the IRB or ethics committee waived the requirement for informed consent. If patients provided informed written consent to have data/samples from their medical records used in research, please include this information.'"

**Additional Editor Comments (if provided):**

This a very large study of HCV genotypes circulating in Brazil.

The study rationale is strong.  However, there are word choice and grammatical issues that require careful proofreading by a native English speaker or a professional editing service.

Please explain further how a real-time PCR assay can test for multiple HCV genotypes in a single sample.

How many samples in this study were genotyped using Sanger sequencing?

What was the lower limit of detection of the real-time PCR assay?

**Comments to the Author**

1. Is the manuscript technically sound, and do the data support the conclusions?

Reviewer #1: Yes

Reviewer #2: Yes

2. Has the statistical analysis been performed appropriately and rigorously? 

Reviewer #1: I Don't Know

Reviewer #2: I Don't Know

3. Have the authors made all data underlying the findings in their manuscript fully available?

Reviewer #1: Yes

Reviewer #2: Yes

4. Is the manuscript presented in an intelligible fashion and written in standard English?

Reviewer #1: No

Reviewer #2: Yes

5. Review Comments to the Author

Reviewer #1: HCV genotype profile in Brazil of mono-infected and HIV co-infected individuals: a survey representative of an entire country

The manuscript has valuable information and is a very good representation of the genotypes distribution throughout Brazil.

1) The text could benefit from English language editing, as some of the sentences' meaning may easily be misinterpreted.

A few examples include:

line 16

HCV genotype from many newly diagnosed Brazilians were determined.

Line 37

Median age of HIV co-infected was 50 years compared to 57 years among mono-infected individuals (71.1% males compared to 58.7%). Distinct prevalence between co-infected and mono-infected individuals were respectively: …

…

Line 240

We also found a high prevalence of HIV co-infection among recently diagnosed HCV-infected individuals (15%), as the co-infection prevalence was higher among men as compared to women.

…

2) It appears that tables 2 and 3 are mislabeled. Please label accordingly and discuss the tables in order of appearance in the text. As is Table 3 is discussed before Table 2.

3) For the table discussing the Median ages by Gender for Groupings of subtypes (currently labeled as 3), please explain in the text the rationale behind the chosen grouping. It is not clear if the data applies to all cases or to the co-infected cases?

Reviewer #2: HCV genotype profile in Brazil...is an interesting paper. However I have some suggestions and critics of the manuscript. There are some topics on the introduction that could be edited. In addition, the results and discussion are limited because of absence of risk factor analysis.

6. PLOS authors have the option to publish the peer review history of their article (what does this mean?). If published, this will include your full peer review and any attached files.

Reviewer #1: No

Reviewer #2: Yes: Maria-Cristina Navas

---

## [Author Response · Author response to Decision Letter 0]

30 Oct 2019

October 30th, 2019

Dear 

Dr. Jason Blackard

Academic Editor

PLOS ONE

We have performed a number of changes to the manuscript PONE-D-19-24177 “HCV genotype profile in Brazil of mono-infected and HIV co-infected individuals: a survey representative of an entire country” to fully address the Reviewer’s and Editor suggestions and concerns. We have included a detailed point-by-point response to the reviewer’s and Editor’s concerns describing the corresponding changes in the manuscript. Please find details bellow, with the responses marked in yellow here.

The Editor requires that: “In ethics statement in the manuscript and in the online submission form, please provide additional information about the patient records/samples used in your retrospective study. Specifically, please ensure that you have discussed whether all data/samples were fully anonymized before you accessed them and/or whether the IRB or ethics committee waived the requirement for informed consent. If patients provided informed written consent to have data/samples from their medical records used in research, please include this information.'

Response to Editor: We therefore have included the following sentences in the Methods section, pages 5-6, lines 116-23 as follows (insertions in yellow here): 

Ethical Statement: This study was approved by the Ethical Committee of the Federal University of São Paulo (#1687/2016). This study was authorized by and performed in partnership with the Sexually Transmitted Infections/AIDS and Viral Hepatitis Department of the Brazilian Ministry of Health. After the initial HCV genotype determination, all data and biological samples from patients were fully anonymized prior to samples being transferred from the Laboratorio Centro de Genomas, São Paulo - Brazil to the Retrovirology Laboratory of the Federal University of Sao Paulo - Brazil for HIV testing and statistical analysis. The Ethical Committee granted this study a waiver of informed consent.

Editor mentions that “We note that you have stated that you will provide repository information for your data at acceptance. 

Response to Editor: This is right, and we included this information on page 7, lines 151-3 (Methods section) as follows: Availability of raw data and files: All relevant data, supporting information files and analytic programs are deposited in a repository entitled HCV Genotype Profile in Brazil hosted at Open Science Framework (https://osf.io/)

The study rationale is strong. However, there are word choice and grammatical issues that require careful proofreading by a native English speaker or a professional editing service.

Response to Editor: The text of the manuscript was carefully revised by two native English speakers from USA that are co-authors in this work, Drs. James Hunter and Hong-Ha M. Truong. We understand that the English was much improved in this version of manuscript. 

Editor asks to “Please explain further how a real-time PCR assay can test for multiple HCV genotypes in a single sample.” 

Response to Editor: In order to clarify this issue, we edited the following sentences in the Methods section, page 6, lines 130-5, as follows: “After an initial PCR amplification using primers designed for HCV conserved region, fluorescent oligonucleotide probes with specific labeling for HCV genetically diverse genomic regions of genotypes 1, 2, 3, 4, 5 and 6, and subtypes 1a and 1b were used for genotype assignment. When more than one distinct fluorescence was detected in a single plasma sample, dedicated software interpreted the result as infections with more than one HCV genotype.” 

Editor enquires: “How many samples in this study were genotyped using Sanger sequencing?”

Response to Editor: 3.2% of samples did not yield a clear genotype assignment using the Abbott Real Time HCV genotype II. As a matter of fact, we have not included in the analysis the HCV genotype results from Sanger sequencing, and we recognize that this information was missing in the manuscript. We are including the information about percentage of samples that have not yielded a clear genotype assignment in lines 135-8 of the methods section as follows. “During the sample collection period, 3.2% of collected plasma samples did not yield a clear genotype assignment using the Abbott Real Time HCV genotype II platform, and these patients were excluded from further analysis.” Also, we removed the sentence mentioning the Sanger sequencing (and former reference 21) as an alternative to obtain genotype results since we are not presenting these in the current manuscript. 

Editor enquires: “What was the lower limit of detection of the real-time PCR assay?” 

Response to Editor: We included this information on lines 129-130 of manuscript in the following sentence (insertion in yellow): “Briefly, the Abbott Real Time HCV genotype II is an in vitro RT-PCR assay using the polymerase chain reaction for the determination of HCV genotypes in the plasma or serum of individuals infected with this virus with the lower detection limit of 500 I.U. of HCV RNA copies/mL.” 

Reviewer #1:

1) The text could benefit from English language editing, as some of the sentences' meaning may easily be misinterpreted.

Reviewer #1 suggested some English language editing as follows: 

line 16

HCV genotype from many newly diagnosed Brazilians were determined.

We edited these sentences as follows (lines 16-7): “We describe the HCV genotype profile of nearly 30,000 individuals in Brazil with recent HCV diagnosis.”

Line 37

Median age of HIV co-infected was 50 years compared to 57 years among mono-infected individuals (71.1% males compared to 58.7%). Distinct prevalence between co-infected and mono-infected individuals were respectively: …

We edited these sentences as follows (lines 38-41): Median age among HCV/HIV co-infected individuals was 50 years old compared to 57 years old among mono-infected individuals. Distinct HCV genotype prevalence between HCV/HIV co-infected and HCV mono-infected individuals were respectively:

Line 240

We also found a high prevalence of HIV co-infection among recently diagnosed HCV-infected individuals (15%), as the co-infection prevalence was higher among men as compared to women.

We edited this sentence as follows (lines 262-4): “We also found a high prevalence of HCV/HIV co-infection among recently diagnosed HCV-infected individuals (15%), and HCV/HIV co-infection prevalence higher among men (18.1%) as compared to women (10.4%).”

2) It appears that tables 2 and 3 are mislabeled. Please label accordingly and discuss the tables in order of appearance in the text. As is Table 3 is discussed before Table 2.

We changed numbering of Tables 2 and 3 in order of appearance in the text and improved the legend for these 2 Tables as follows: 

Table 2 (former Table 3): Difference in mean and median ages by gender according to HCV genotype groupings.

Table 3 (former Table 2). HCV genotype prevalence among the total number of individuals, HCV mono-infected individuals and HCV/HIV co-infected individuals according to gender and Brazilian geographic regions. 

3) Reviewer 2 mentions that “For the table discussing the Median ages by Gender for Groupings of subtypes (currently labeled as 3), please explain in the text the rationale behind the chosen grouping. It is not clear if the data applies to all cases or to the co-infected cases?”

Response to reviewer 2: In order to clarify that, we included the following sentence on page 8, lines 175- 8 as follows: “We also grouped the subtypes into those with higher and lower median ages. Among these, the mean age was significantly lower among individuals infected with G1A, G3 and G4 for both genders (p < 10-6), as shown in Table 2.”

Reviewer 2 observes that “There are some topics on the introduction that could be edited.”, and we believe we have done it. 

Reviewer 2 observes that “the results and discussion are limited because of absence of risk factor analysis.” 

We agree with the reviewer, and although available, gathering this specific information would jeopardize the anonymous nature of the study, raising ethical concerns. We therefore mention that as a limitation of the study, and included the following sentence at lines 307-309 of the Discussion: “We also recognize that the lack of data on risk factors for HCV and HIV acquisition precludes more definite conclusions and speculations about the HCV genotype profiles described here.” 

We sincerely believe that our manuscript is now improved as a result of recommendations. We look forward to hearing back from you regarding our revised manuscript.

Ricardo Sobhie Diaz, M.D., PhD.

Retrovirology Laboratory

Federal University of Sao Paulo, Infectious Diseases Department 

R. Pedro de Toledo, 781, Sao Paulo, SP - 04039 - Brazil

Phone: (55-110) 9109-0445/ (55-11) 5084-4262

Email: rsdiaz@catg.com.br

---

## [Decision Letter · Decision Letter 1]

12 Dec 2019

HCV genotype profile in Brazil of mono-infected and HIV co-infected individuals: a survey representative of an entire country

PONE-D-19-24177R1

Dear Dr. Diaz,

We are pleased to inform you that your manuscript has been judged scientifically suitable for publication and will be formally accepted for publication once it complies with all outstanding technical requirements.

With kind regards,

Jason Blackard, PhD

Academic Editor

PLOS ONE

Additional Editor Comments (optional):

None

Reviewers' comments:

Reviewer's Responses to Questions

**Comments to the Author**

1. If the authors have adequately addressed your comments raised in a previous round of review and you feel that this manuscript is now acceptable for publication, you may indicate that here to bypass the “Comments to the Author” section, enter your conflict of interest statement in the “Confidential to Editor” section, and submit your "Accept" recommendation.

Reviewer #1: All comments have been addressed

Reviewer #2: All comments have been addressed

2. Is the manuscript technically sound, and do the data support the conclusions?

Reviewer #1: Yes

Reviewer #2: Yes

3. Has the statistical analysis been performed appropriately and rigorously? 

Reviewer #1: Yes

Reviewer #2: N/A

4. Have the authors made all data underlying the findings in their manuscript fully available?

Reviewer #1: Yes

Reviewer #2: Yes

5. Is the manuscript presented in an intelligible fashion and written in standard English?

Reviewer #1: Yes

Reviewer #2: Yes

6. Review Comments to the Author

Reviewer #1: PONE-D-19-24177R1

HCV genotype profile in Brazil of mono-infected and HIV co-infected individuals: a survey representative of an entire country. The manuscript is valuable and has good representation of the genotypes distribution throughout Brazil.

The reviewer’s comments have been addressed carefully and thoroughly. The manuscript now appears ready for publication

Reviewer #2: (No Response)

7. PLOS authors have the option to publish the peer review history of their article (what does this mean?). If published, this will include your full peer review and any attached files.

Reviewer #1: No

Reviewer #2: No

---

## [Editor Report · Acceptance letter]

26 Dec 2019

PONE-D-19-24177R1 

HCV genotype profile in Brazil of mono-infected and HIV co-infected individuals: a survey representative of an entire country 

Dear Dr. Diaz:

I am pleased to inform you that your manuscript has been deemed suitable for publication in PLOS ONE. Congratulations! Your manuscript is now with our production department. 

With kind regards,

on behalf of

Dr. Jason Blackard 

Academic Editor

PLOS ONE